# Does Representation Similarity Capture Function Similarity?

**Lucas Hayne** *lucas.hayne@colorado.edu*
*University of Colorado Boulder*

**Heejung Jung** *heejung.jung@colorado.edu*
*University of Colorado Boulder*

**R. McKell Carter** *mckell.carter@colorado.edu*
*University of Colorado Boulder*

**Reviewed on OpenReview:** *https://openreview.net/forum?id=YY2iAOhfia*

## Abstract

Representation similarity metrics are widely used to compare learned representations in neural networks, as is evident in extensive literature investigating metrics that accurately capture information encoded in representations. However, aiming to capture all of the information available in representations may have little to do with what information is actually used by the downstream network. One solution is to experiment with interventions on network function. By ablating groups of units thought to carry information and observing whether those ablations affect network performance, we can focus on an outcome that mechanistically links representations to function.

In this paper, we systematically test representation similarity metrics to evaluate their sensitivity to functional changes induced by ablation. We use network performance changes after ablation as a way to measure the influence of representation on function. These measures of function allow us to test how well similarity metrics capture changes in network performance versus changes to linear decodability. Network performance measures index the information used by the downstream network, while linear decoding methods index available information in the representation.

We show that all of the tested metrics are more sensitive to decodable features than network performance. When comparing these metrics, Procrustes and CKA outperform regularized CCA-based methods on average. Although Procrustes and CKA outperform on average, these metrics have a diminished advantage when looking at network performance. We provide ablation tests of the utility of different representational similarity metrics. Our results suggest that interpretability methods will be more effective if they are based on representational similarity metrics that have been evaluated using ablation tests.

## 1 Introduction

Neural networks already play a critical role in systems where understanding and interpretation are paramount like in self-driving cars and the criminal justice system. To understand and interpret neural networks, learned representations are compared using representation similarity metrics (RSMs) (Kornblith et al. (2019); Raghu et al. (2017); Morcos et al. (2018b); Wang et al. (2018); Li et al. (2015); Feng et al. (2020); Nguyen et al. (2020)). Using these similarity metrics, researchers evaluate whether networks trained from different random initializations learn the same information, whether different layers learn redundant or complementary information, and how different training data affect learning (Kornblith et al. (2019); Li et al. (2015); Wang et al. (2018)). Furthermore, RSMs are also used to compare computational models and biological networks (Kriegeskorte et al. (2008)) to determine the ability of networks to model brains.

In a prototypical example comparing neural systems, Raghu et al. (2021) used RSMs to help answer the question: "Do vision transformers see like convolutional neural networks?" Put another way, the researchers aim to answer whether the two networks function in a similar way. Two networks can be shown to function in a similar way when two criteria are met. First, the two networks must compute similar representations. What it means for two representations to be similar, however, is not straightforward. In response to this vagueness, many RSMs have been developed with different notions of similarity over the past decade. Different similarity metrics employ different strategies and make different assumptions, meaning that different similarity metrics might not always agree. For example, some similarity metrics are invariant under invertible linear transformations while others are not (see Kornblith et al. (2019) for a theoretical comparison). Capturing the various notions of representational similarity has been the primary driver behind developing and testing RSMs. These different assumptions and strategies can lead to quantitatively different predictions. For instance, Davari et al. (2022) demonstrate that the centered kernel alignment metric predicts a high similarity between random and fully trained representations. It is unclear which representation similarity metrics capture the most important information from representations, further tests are needed to evaluate them.

However, to address the original question: whether two networks function similarly, an additional second criteria must be met. Apart from computing similar representations, the two networks must also use their representations in a similar way. This second criterion has been mostly neglected when considering the similarity of network function using RSMs. In this paper, we aim to investigate whether RSMs can tell us when this second criterion is met. In other words, how likely is it that two representations deemed similar by RSMs are actually used the same way by trained networks?

In this work, we aim to quantify how likely is it that two representations deemed similar by RSMs are actually used the same way by trained networks. Previous studies have tackled this question by making assumptions about what kinds of information networks use. If this information is detectable using RSMs, then RSMs can be used to determine if two representations are used by the network in the same way. For instance, RSMs have been evaluated for their ability to track class separability in representations (Boix-Adsera et al. (2022); Ding et al. (2021); Feng et al. (2020)). To measure the class separability in a representation, researchers usually train linear probes for downstream tasks on learned representations and compare the results. However, the features of a representation that carry the most information for linear probes may not be those actually used by the network during inference. Studies that remove features from representations in trained networks have revealed a weak link between the relevance of a feature for decoding and its effect when removed from the network (Meyes et al. (2020); Zhou et al. (2018); Donnelly & Roegiest (2019); Morcos et al. (2018b)). Hayne et al. (2022) recently showed that linear decoders specifically cannot single out the features of representations actually used by the network. Consequently, two representations that are equally decodable using linear probes may not actually be equal from the point of view of network performance.

If networks neglect linearly decodable information, then RSMs that track linear class separability may not satisfy the second criterion mentioned previously. Remember the second criterion says that to infer the functional similarity of two networks from RSM scores, one must also show that networks use similar representations in similar ways. In this work, we propose using ablations to directly test if RSMs satisfy the second criterion. We use ablation to evaluate how closely changes in RSM scores are related to changes in network performance. We first ablate groups of units from either AlexNet, MobileNet, or ResNet, compare the original representations to the ablated representations using RSMs, and then compare scores to the changes seen for network performance (see Figure 1). As a baseline, we also compare RSM scores to changes in class separability in the representation using linear probes. This process reveals the extent to which RSMs satisfy the second criterion, tracking functional similarities as opposed to just linear similarities between representations. This distinction is crucial for neural network interpretability where the aim is to develop human-understandable descriptions of how neural networks actually rely on their internal representations.

In this work, we show that CKA, Procrustes, and regularized CCA-based representation similarity metrics predict network performance changes significantly worse than linear decoding changes. We also show that, on average, Procrustes and CKA outperform regularized CCA-based methods. The advantage of using Procrustes and CKA is significantly diminished, however, when considering network performance tests. In fact, Procrustes and CKA fail to outperform regularized CCA-based metrics in AlexNet. Overall, our results

suggest that interpretability methods will be more effective if they are based on representational similarity metrics that have been evaluated using ablation tests. In general, this paper documents the following contributions:

- We introduce a new test of the utility of representation similarity metrics. We find that five popular representation similarity metrics are significantly less sensitive to network performance changes induced by ablation than linearly decodable changes.

- Within the tested metrics, we show that Procrustes and CKA tend to outperform regularized CCA-based methods for capturing functional similarities between representations, but that tests using linear probes and network performance based functional measures can produce different results in different networks.

## 2 Methods

Here, we described the methods necessary for correlating representational metrics with changes in decodability and ablation effects. In Section 2.1, we describe the statistical testing methodology used in our experiments. In Section 2.2, we introduce the representation similarity measures we evaluate and reformulate them for use on high dimensional representations. In Section 2.3, we describe how we use ablation to produce representations with different functional properties. Finally, in Section 2.4, we describe how we use linear probe decoding deficits and class-specific performance deficits to measure decodable and downstream network changes, respectively, in the ablated representations.

### 2.1 Statistical testing

Assume $A \in \mathbb{R}^{n \times p_1}$ represents a matrix of activations for $p_1$ neurons given $n$ examples, and $B \in \mathbb{R}^{n \times p_2}$ represents a matrix of activations for $p_2$ neurons given the same $n$ examples. The matrices $A$ and $B$ are called representation matrices and have been preprocessed to have centered columns. Let $RSM(A, B)$ denote a representation similarity metric that returns zero if and only if $A = B$ and for which $RSM(A, B) = RSM(B, A)$. These metrics do not satisfy the triangle inequality and are therefore not formal distance metrics. For simplicity, we will refer to them as metrics in this work.

Separate from representational differences, we also seek to quantify how changes in representation affect function. To quantify functional differences between representation matrices, we use functional behavior measures. Formally, let $f : \mathbb{R}^{n \times p} \to \mathbb{R}$ denote a functional behavior measure that, given a representation matrix, returns a scalar measure of the representation's role in function. In this study, we utilize two functionality measures, class-specific linear decoding accuracy ($f_{\mathrm{Dec}}$) and class-specific network performance ($f_{\mathrm{Perf}}$). In the case of linear decoding accuracy, $f_{\mathrm{Dec}}$ returns the average linear probe accuracy achieved from decoding a target class identity from a representation matrix. On the other hand, $f_{\mathrm{Perf}}$ returns the average classification performance for a target class achieved by feeding the representation matrix to the network at the appropriate layer. More details are presented in Section 2.4.

To change the representations themselves, we perform interventions on a given representation $Z$ (Pearl (2009)). Interventions on $Z$ come in the form of zero ablations where entire features (columns of $Z$) are set to zero using a mask producing an ablated representation $A$ where $A = M \odot Z$. The functionality of the masked representation is evaluated using a linear decoder ($f_{\mathrm{Dec}}$) and the downstream neural network ($f_{\mathrm{Perf}}$). We represent the zero ablation intervention on functionality with the $\mathrm{do}(\cdot)$ operator. For each image $x$, we compute the predicted target class rank $r_t(x)$ for the target class $t$. For instance, if the CNN's softmax layer predicts that the Junco bird class was the third most likely class given an image of a Junco bird, then that image receives a rank of three ($r_t(x) = 3$). The impact of the ablation on function is registered as the difference in predicted target class ranks produced by the intervention on the representation.

$$\Delta_{\mathrm{ablate}} = [r_t(x) - r(x|\mathrm{do}(A))] \tag{1}$$

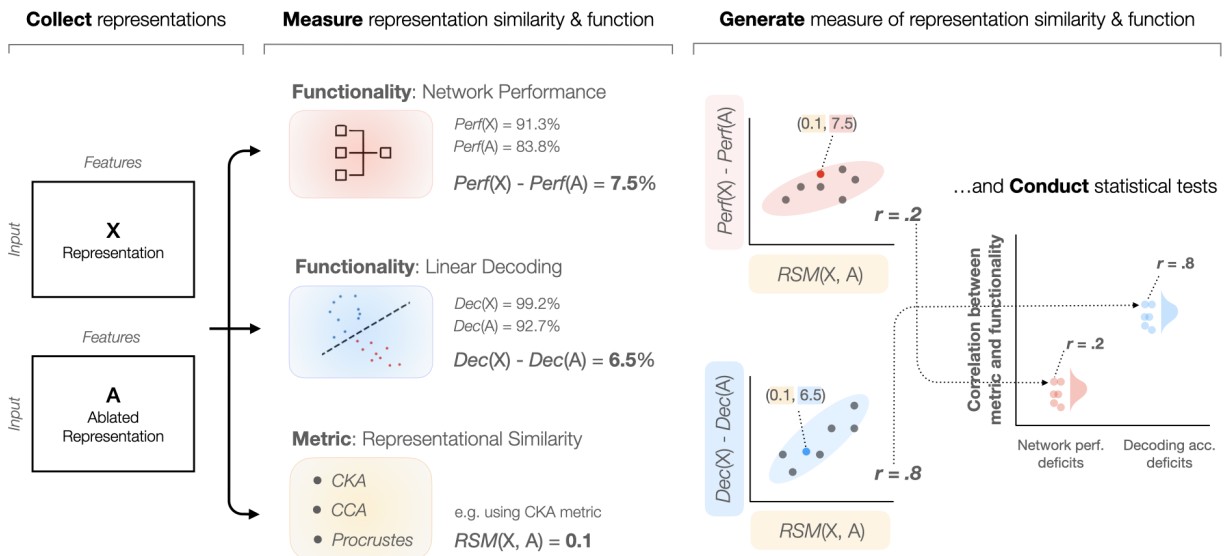

Figure 1: **Elements of the experimental design.** In the first phase, a representation matrix $\boldsymbol{X}$ is extracted from a layer in a trained network and we then generate ablation matrices $\boldsymbol{A}$ by deleting features from the representation. In the second phase, representation matrices $\boldsymbol{X}$ and $\boldsymbol{A}$ are compared in three ways. To measure how the representations differ in the context of the network, the representations are fed back into the network and the network performance difference between them is calculated. Next, linear probes are also fit to $\boldsymbol{X}$ and $\boldsymbol{A}$ to decode a target class and the linear probe accuracies are compared. Additionally, a representation metric similarity is calculated between $\boldsymbol{X}$ and $\boldsymbol{A}$. In the last phase, representation similarity and function are compared. By comparing metric similarities with both linear probe decoding changes and network performance changes across many ablations (represented with multiple points in the correlation plots), we can assess to what extent each metric can be used to infer functional similarity from representation similarity or whether they merely capture changes in class separability. Finally, statistical tests are performed on the correlation values in each experiment.

Similarly, for linear decoders, the impact of ablation on class separability is registered as the drop in linear decoder accuracy after retraining on the ablated representation (see Section 2.4 for more details).

As in Ding et al. (2021), we aim to statistically test representation similarity metrics using a similar methodology according to their paper:

1. Extract a representation matrix $\boldsymbol{X} \in \mathbb{R}^{n \times p_{\text{input}}}$ from a given layer in a neural network.

2. Collect a set of representations $\mathbb{A}$ by masking groups of neurons in $\boldsymbol{X}$ to produce $\boldsymbol{A}_i \in \mathbb{R}^{n \times p_{\text{ablated}}}$, where $p_{\text{ablated}} < p_{\text{input}}$ (see Section 2.3).

3. Compute the following for all $\boldsymbol{A}_i \in \mathbb{A}$ :

   - $D_i = RSM(\boldsymbol{X}, \boldsymbol{A}_i)$
   - $F_{Perf}(\boldsymbol{X}, \boldsymbol{A}_i) = |f_{Perf}(\boldsymbol{X}) - f_{Perf}(\boldsymbol{A}_i)|$
   - $F_{Dec}(\boldsymbol{X}, \boldsymbol{A}_i) = |f_{Dec}(\boldsymbol{X}) - f_{Dec}(\boldsymbol{A}_i)|$

4. Compute the Spearman rank correlation between $D$ and $F_{Perf}$ and $D$ and $F_{Dec}$.

This procedure quantifies the extent to which the representation similarity metrics, $RSM(\cdot)$, capture the functionality differences, as measured by $f_{Perf}(\cdot)$ and $f_{Dec}(\cdot)$, produced by ablating the representation matrix. A high Spearman's rank correlation value between a metric's computed representation similarities and the functionality differences produced by ablation implies that the chosen metric is *sensitive* to the chosen functionality. Whereas, a low correlation implies the opposite: that the chosen metric is not sensitive to the chosen functionality.

## 2.2 Representation similarity metrics

As in Ding et al. (2021), we study three main representation similarity metrics: centered kernel alignment (CKA), Procrustes, and canonical correlation analysis (CCA).

**Centered kernel alignment (CKA)** is based on the idea that similar representations also have similar relations between examples. In other words, representation matrices that store images of lettuce and rabbits using similar vectors should be more similar to each other than with representation matrices that encode images of lettuce and dinner plates using similar vectors. This idea leads Kornblith et al. (2019) to formulate linear CKA, which uses a linear kernel to compare example vectors (henceforth referred to as just CKA):

$$d_{\mathrm{CKA}}(\boldsymbol{A}, \boldsymbol{B}) = 1 - \frac{\|\boldsymbol{A}^{\mathrm{T}}\boldsymbol{B}\|_{\mathrm{F}}^2}{\|\boldsymbol{A}^{\mathrm{T}}\boldsymbol{A}\|_{\mathrm{F}}\|\boldsymbol{B}^{\mathrm{T}}\boldsymbol{B}\|_{\mathrm{F}}} \tag{2}$$

where $\|\cdot\|_{\mathrm{F}}$ is the Frobenius norm and $\|\boldsymbol{A}^{\mathrm{T}}\boldsymbol{B}\|_{\mathrm{F}}^2$ derives from the following relation:

$$\langle \mathrm{vec}(\boldsymbol{A}\boldsymbol{A}^{\mathrm{T}}), \mathrm{vec}(\boldsymbol{B}\boldsymbol{B}^{\mathrm{T}})\rangle = \mathrm{tr}(\boldsymbol{A}\boldsymbol{A}^{\mathrm{T}}\boldsymbol{B}\boldsymbol{B}^{\mathrm{T}}) = \|\boldsymbol{A}^{\mathrm{T}}\boldsymbol{B}\|_{\mathrm{F}}^2 \tag{3}$$

Relation 3 (which was derived by Kornblith et al. (2019)) shows that the similarity between pairwise example similarity matrices (far left) is equal to the squared Frobenius norm of the feature covariance matrix between representations (far right). Kornblith et al. (2019) use this relation to form Equation 2 which measures the normalized similarity between the example similarity matrices of $\boldsymbol{A}$ and $\boldsymbol{B}$. Unfortunately, computing and storing either $\boldsymbol{A}^{\mathrm{T}}\boldsymbol{B}$, $\boldsymbol{A}^{\mathrm{T}}\boldsymbol{A}$, or $\boldsymbol{B}^{\mathrm{T}}\boldsymbol{B}$ can be prohibitively expensive when both $p_1$ and $p_2$ grow too large. Therefore we reformulate Equation 2 using relation 3 from Kornblith et al. (2019) and the fact that $\|\boldsymbol{X}^{\mathrm{T}}\boldsymbol{X}\|_{\mathrm{F}} = \|\boldsymbol{X}\boldsymbol{X}^{\mathrm{T}}\|_{\mathrm{F}}$ into:

$$d_{\mathrm{CKA}}(\boldsymbol{A}, \boldsymbol{B}) = 1 - \frac{trace(\boldsymbol{A}\boldsymbol{A}^{\mathrm{T}}\boldsymbol{B}\boldsymbol{B}^{\mathrm{T}})}{\|\boldsymbol{A}\boldsymbol{A}^{\mathrm{T}}\|_{\mathrm{F}}\|\boldsymbol{B}\boldsymbol{B}^{\mathrm{T}}\|_{\mathrm{F}}} \tag{4}$$

This reformulation allows us to use CKA on layers with $n \ll p_1, p_2$.

**Procrustes** is an analytical solution to the orthogonal Procrustes problem which involves finding a right rotation of matrix $\boldsymbol{B}$ that is as close as possible to $\boldsymbol{A}$ as measured by the Frobenius norm:

$$d_{\mathrm{Procrustes}}(\boldsymbol{A}, \boldsymbol{B}) = \|\boldsymbol{A}\|_{\mathrm{F}}^2 + \|\boldsymbol{B}\|_{\mathrm{F}}^2 - 2\|\boldsymbol{A}^{\mathrm{T}}\boldsymbol{B}\|_* \tag{5}$$

where $\|\cdot\|_*$ is the nuclear norm. As with CKA, $\boldsymbol{A}^{\mathrm{T}}\boldsymbol{B}$ needs to be replaced to lighten the computational cost of working with large layers. Therefore, we utilize the fact that the nuclear norm of a matrix is the sum of its singular values to reformulate Procrustes:

$$d_{\mathrm{Procrustes}}(\boldsymbol{A}, \boldsymbol{B}) = \|\boldsymbol{A}\|_{\mathrm{F}}^2 + \|\boldsymbol{B}\|_{\mathrm{F}}^2 - 2\sum_{i}^{n} \sqrt{\lambda_i(\boldsymbol{A}\boldsymbol{A}^{\mathrm{T}}\boldsymbol{B}\boldsymbol{B}^{\mathrm{T}})} \tag{6}$$

where $\lambda_i(\boldsymbol{X})$ represents the $i^{\mathrm{th}}$ eigenvalue of matrix $\boldsymbol{X}$. Again, this reformulation saves us from manipulating the large $p_1 \times p_2$ matrix by replacing it with a much smaller $n \times n$ matrix (assuming $n \ll p_1, p_2$).

**Canonical correlation analysis (CCA)** provides a solution to the problem of linearly projecting $\boldsymbol{A}$ and $\boldsymbol{B}$ into a shared subspace where their correlations are maximized. CCA finds $\min(p_1, p_2)$ pairs of weight

vectors $(\boldsymbol{w_A}, \boldsymbol{w_B})$ and the resulting correlation induced by projecting $\boldsymbol{A}$ and $\boldsymbol{B}$ using the $i^{\text{th}}$ weight vector is:

$$\rho_i(\boldsymbol{A}, \boldsymbol{B}) = \max_{\boldsymbol{w_A^i}, \boldsymbol{w_B^i}} \text{corr}(\boldsymbol{A}\boldsymbol{w_A^i}, \boldsymbol{B}\boldsymbol{w_B^i}) \quad \text{s.t.} \quad \forall_{j<i} \quad \boldsymbol{A}\boldsymbol{w_A^i} \perp \boldsymbol{A}\boldsymbol{w_A^j}, \quad \boldsymbol{B}\boldsymbol{w_B^i} \perp \boldsymbol{B}\boldsymbol{w_B^j} \tag{7}$$

where the $\rho_i$ is maximized subject to the constraint that the subspace features be orthogonal. Equation 7 can be solved for by performing singular value decomposition on $(\boldsymbol{A}^{\text{T}}\boldsymbol{A})^{-1/2}\boldsymbol{A}^{\text{T}}\boldsymbol{B}(\boldsymbol{B}^{\text{T}}\boldsymbol{B})^{-1/2}$ where the singular values are equal to the correlations ($\rho_i \, \forall \, i \in [1, ..., \min(p_1, p_2)]$).

However, the inverses of the feature covariance matrices do not exist when the number of neurons exceeds the number of examples. For these cases, we can use regularized or "ridge" CCA (Vinod (1976)), which applies an $L_2$ penalty to the weight vectors and can be solved by performing SVD on $(\boldsymbol{A}^{\text{T}}\boldsymbol{A} + \kappa_{\boldsymbol{A}}\boldsymbol{I})^{-1/2}\boldsymbol{A}^{\text{T}}\boldsymbol{B}(\boldsymbol{B}^{\text{T}}\boldsymbol{B} + \kappa_{\boldsymbol{B}}\boldsymbol{I})^{-1/2}$. However, we again run into the problem that the feature covariance matrices are too costly to compute for large layers. So, we employ the "kernel trick" introduced by Kuss & Graepel (2003) and Hardoon et al. (2004) and refined by Tuzhilina et al. (2021) which allows us to substitute into the above expression $\boldsymbol{R_A}$ for $\boldsymbol{A}$ where $\boldsymbol{R_A}$ represents an $n \times n$ matrix recovered by applying SVD on $\boldsymbol{A}$, i.e. $\boldsymbol{A} = \boldsymbol{R_A}\boldsymbol{V_A^{\text{T}}}$. The same trick can be applied to $\boldsymbol{B}$. The kernel trick (Tuzhilina et al. (2021)) makes CCA computationally tractable for large layers. The only caveat is that if we apply the "kernel trick" to both matrices, we recover only $n$ canonical correlations rather than $\min(p_1, p_2)$.

Mean CCA (as used by Raghu et al. (2017)) and mean squared CCA (Ramsay et al. (1984)) average raw and squared correlations recovered through CCA, respectively. Projection-weighted canonical correlation analysis (PWCCA) is a special case of canonical correlation analysis (CCA) proposed by Morcos et al. (2018a). PWCCA re-weights each correlation value by its importance for the underlying representation. Formally, if representation matrix $\boldsymbol{A}$ has neuron activation vectors $[\boldsymbol{z}_1, ..., \boldsymbol{z}_{p_1}]$ and CCA vectors $[\boldsymbol{h}_1, ..., \boldsymbol{h}_n]$, then PWCCA computes a weighted mean as:

$$d_{\text{PWCCA}}(\boldsymbol{A}, \boldsymbol{B}) = 1 - \sum_{i=1}^{n} \tilde{\alpha}_i \rho_i \quad \text{s.t.} \quad \alpha_i = \sum_j |\langle \boldsymbol{h}_i, \boldsymbol{z}_j \rangle|$$

where $\tilde{\alpha}_i$ is a normalized version of $\alpha_i$. The preceding CCA reformulations represent regularized forms of CCA-based metrics useful for high-dimensional representations, which we refer to as regularized CCA-based metrics in the text, but just CCA metrics in the figures for sake of brevity.

There is some concern that the regularized approximation could introduce errors into our results that interfere with the conclusions of the paper. Observing different penalties for regularizing the CCA metrics, we find that the exact CCA values do change when the penalties change. However, in the current study, we are not concerned with the exact CCA values, only the relative order of scores given an ablation. We find that CCA metrics regularized with different penalties are highly correlated (Spearman rank correlation > 0.95) and therefore the chosen penalty does not interfere with the main results.

## 2.3 Ablation

To study representation functionality changes, we needed a method for manipulating representations to reliably produce network performance deficits at the output of the network. To reliably produce network performance deficits, we followed the procedure of Hayne et al. (2022). Specifically, for each of the 10 or 50 randomly chosen target classes and layer of the CNNs we tested, AlexNet, MobileNetV2, and ResNet50, we first projected every neuron onto two dimensions: class selectivity and activation magnitude. Then, we constructed a grid to overlay on the activation space so that each cell of the grid contained the same number of neurons. To supply the set $A$ of representation matrices from Section 2.1, we ablated one cell of neurons at a time by setting the activation values for those neurons to zero.

## 2.4 Functionality measures

After collecting the set $\mathbb{A}$ of ablated representation matrices, we sought to compare two functionality measures. First, we fit linear probe decoders to each representation matrix in $\mathbb{A}$. With the linear probes we aimed to decode the identity of one target class, so we fit simple logistic regression classifiers to distinguish the target class representations from all other representations in the representation matrix. Because many of the CNN layers contained thousands of features, we randomly selected 100 neurons as features in the logistic regression model and averaged the training accuracy over 200 repetitions as in Alain & Bengio (2016). Intuitively, the accuracies measure how linearly separable the target class is from other classes on average given the features in a representation. We refer to the changes induced in the decodability of the target class by ablation as decoding accuracy deficits.

In contrast to linear decoding accuracies, we also measured the network's class-specific classification performance when utilizing the representations in the representation matrix during inference. Specifically, we recorded the average classification rank of the target class. For instance, if the CNN's softmax layer predicted that the Junco bird class was the third most likely class given an image of a Junco bird, then that image received a rank of three. The average class-specific ranks intuitively measure how the representation was used by the network. We refer to the changes induced in class-specific classification rank by ablation as network performance deficits.

## 2.5 Networks and data

For our experiments, we investigated AlexNet (Krizhevsky et al. (2012)), MobileNetV2 (Sandler et al. (2018)), and ResNet50 (He et al. (2016)) all pre-trained on ImageNet (Deng et al. (2009)) [1]. We chose these three networks as a representative sample of three convolutional networks with different architectures that are still widely studied for producing, testing, and analyzing representation similarity metrics (Kornblith et al. (2019); Raghu et al. (2021); Boix-Adsera et al. (2022)). AlexNet and MobileNetV2 were built using Keras and included lambda masking layers after each parameterized layer to selectively ablate unit groups. ResNet50 was built using Pytorch with forward hooks applied to the output of each block and used to perform ablation. For MobileNetV2 and ResNet50, 10 classes were randomly chosen from ImageNet and all the images from the validation set were used for our analysis. For AlexNet, 50 random classes were chosen. We choose 50 classes for AlexNet and 10 classes for MobileNet and ResNet to strike a balance between convenience and breadth. Given the large layer sizes of MobileNet and ResNet ($\sim$100,000 neurons), adding more classes increases the cost of computing RSM scores. To help avoid bias in our small sample of classes, we sample the classes completely randomly from ImageNet. We note that previous studies include a comparable number of images in their analysis (Raghu et al. (2021)).

# 3 Results

In this section, we detail the results of aggregating our statistical tests across each layer and class from AlexNet, MobileNetV2, and ResNet50. We conducted a hierarchical linear model to evaluate the prediction of the rank correlation values as a function of two factors: 1) *metric*, i.e. CKA, Procrustes, and regularized CCA-based methods and 2) *functionality*, i.e. network performance deficits and decoding accuracy deficits. To allow for comparable correlation coefficients, we Fisher-Z transformed each rank correlation value. This transformation normalizes the distribution of coefficients, which makes it suitable for averaging and executing further statistical analyses. After Fisher-Z transformation, we implement a hierarchical linear model as a function of functionality and metric, while allowing for random intercepts and random slope of functionality. The model is as follows:

---

[1] AlexNet pre-trained weights from `http://github.com/heuritech/convnets-keras`. MobileNetV2 weights were downloaded from `http://keras.io/api/applications`. ResNet50 weights were downloaded from `https://pytorch.org/hub/pytorch_vision_resnet/`

$$\text{FisherZ}_{ij} = \beta_0 + \beta_1 \times \text{Functionality}_{ij} + \beta_2 \times \text{Metric}_{ij} + \beta_3 \times (\text{Functionality}_{ij} \times \text{Metric}_{ij})$$
$$+ u_{0j} + u_{1j} \times \text{Functionality}_{ij} + R_{ij} \tag{8}$$

$$\begin{bmatrix} u_{0j} \\ u_{1j} \end{bmatrix} \sim N \left( \begin{bmatrix} 0 \\ 0 \end{bmatrix}, \begin{bmatrix} \tau_{0j}^2 & \rho_{0j,1j} \times \tau_{0j} \times \tau_{1j} \\ \rho_{1j,0j} \times \tau_{1j} \times \tau_{0j} & \tau_{1j}^2 \end{bmatrix} \right) \tag{9}$$

$$R_{ij} \sim N(0, \sigma^2) \tag{10}$$

***Class*** $j$: the unit of analysis, also referred to as clusters. Given that observations from the same cluster are bound to be non-independent, we model classes as our unit of analysis. ***FisherZ***$_{ij}$: the dependent variable for the $i$-th observation in the $j$-th class. ***Functionality***$_{ij}$: the contrast code to represent different levels of functionality for the $i$-th observation in the $j$-th class. ***Metric***$_{ij}$: the contrast codes to represent levels of metrics – CKA, Procrustes, and regularized CCA-based methods of mean CCA, mean squared CCA, and PWCCA – for the $i$-th observation in the $j$-th class. $\boldsymbol{\beta_0}$, $\boldsymbol{\beta_1}$, $\boldsymbol{\beta_2}$, $\boldsymbol{\beta_3}$: the fixed effect coefficients for the intercept, Functionality, Metric, and their interaction respectively. For instance, $\beta_1$ represents the average difference between Network performance deficits and Decoding accuracy deficits across classes. $\boldsymbol{u_{0j}}$: the random intercept for the $j$-th class. $\boldsymbol{u_{1j}}$: the random slope for the Functionality factor, for the $j$-th class. In other words, we allow for the functionality effects to vary across classes, e.g., the average difference between Network performance deficits and Decoding accuracy deficits can differ across classes. The random effects vector $\begin{bmatrix} \boldsymbol{u_{0j}} & \boldsymbol{u_{1j}} \end{bmatrix}^\top$ is a multivariate normal distribution with a mean of 0 respectively and a covariance structure with 1) $\boldsymbol{\tau_{0j}^2}$, the variance for the random intercepts ($u_{0j}$), 2) $\boldsymbol{\tau_{1j}^2}$, the variance for the random slopes ($u_{1j}$), and 3) the covariance of the random intercepts and slopes, indicated as the correlation between the random slopes and the random intercepts $\boldsymbol{\rho_{1j,0j}}$ multiplied with the standard deviation of the random intercepts $\boldsymbol{\tau_{0j}}$ and random slopes $\boldsymbol{\tau_{1j}}$. $\boldsymbol{R}_{ij}$: the residual error term for each observation, after fitting the model with aforementioned parameters.

For visualization, we plot the raw correlation coefficients, for better interpretability. We average the correlation coefficients within classes, i.e., the unit of analyses, across layers for each each metric and functionality. Each data point represents a unique class, plotted as a function of the factor of interest. Distribution of the class-average correlation coefficients are plotted to visualize the spread of the data per factor. Boxplots indicate the median, and the 25th and 75th percentile of the class data distribution (i.e. inter-quartile range; IQR). The black whiskers represent $1.5 \times$ IQR.

### 3.1 Representation similarity metrics are significantly less sensitive to network function

Figure 2 shows the distribution of correlation values across functionality measures. In all cases, the rank correlation values of both functionality measures are significantly different from 0, suggesting that the correlation values are sensitive to functional behavioral changes in the representations. The analysis of interest demonstrates these correlation values are significantly different between network performance deficits and decoding accuracy deficits, when averaging across the five similarity metrics — CKA, Procrustes, and regularized CCA-based metrics. In other words, there is a significant main effect of functionality. The distribution of Spearman correlation values between the metrics and each functionality for each network are shown in Figure 2.

### 3.2 CKA and Procrustes tend to outperform other metrics

Figure 3 shows that CKA and Procrustes have, on average, higher rank correlation values compared to the other CCA-based metrics. Again, higher rank correlation values indicate a higher coupling between functionality measures and representation similarity metrics, thereby suggesting that CKA and Procrustes are better metrics at capturing functionality in general.

### 3.3 Functionalities produce different results for different networks

Figure 4 (left) shows a significant interaction between functionality measures and similarity metrics for AlexNet. Figure 4 (middle) shows the same model for MobileNetV2, however, interaction is non-significant. Figure 4 (right) shows the conceptually identical interaction, which is significant for ResNet.

In the case of AlexNet, with the plotted means, it is evident that the interaction is driven by the metric differences within decoding accuracy deficits. As for the similarity metrics within network performance deficits, no single metric outperforms the other; instead, each metric achieves a similarly low correlation amongst the network performance deficit group. As for the decoding accuracy deficit group, on the other hand, CKA and Procrustes outperform the CCA-based metrics. In the case of MobileNetV2 and ResNet, there is a significant interaction where the difference between CKA/Procrustes and the CCA-based metrics for network performance tests is significantly different from decoding tests.

## 4 Discussion

In this work, we systematically test representation similarity metrics on ImageNet-trained CNNs to determine how sensitive they are to network performance and decoding changes in representations. These tests help us answer: how likely is it that two representations deemed similar by RSMs are actually used the same way by trained networks? Our tests revealed that, while similarity metrics are significantly sensitive to network performance and decoding measures of functionality, they are significantly less sensitive to network performance. In other words, RSM scores are somewhat predictive of what ablations will have the largest effects on network function, but are comparatively much better at predicting changes in class separability due to ablation. Additionally, in most of our tests CKA and Procrustes significantly outperform CCA-based methods at predicting functional changes. However, we do find that tests of network function tend to significantly reduce and, in the case of AlexNet, eliminate their advantages.

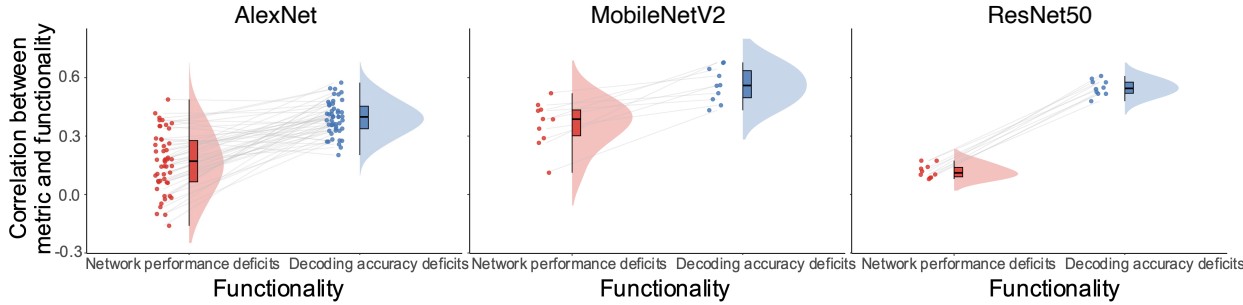

Figure 2: **Correlation values overall are significantly sensitive to ablation changes in the representations, however, decoding accuracy has a higher value than network performance deficits.** This figure shows the distribution of rank correlation values between functionality measures, i.e., the main effect of functionality measure on AlexNet (a), MobileNetV2 (b), and ResNet50 (c). Each data point represents a value from the ten classes; within each class, we average the five metrics of CKA, Procrustes, PWCCA, mean CCA and mean squared CCA. The distribution is further depicted by the violin plots and boxplots, which illustrates the median and upper/lower quartile of the distribution. Overall, both functionality measures are significantly different from 0, indicating that the rank correlation values are sensitive to changes produced by ablations in the representations ($F(1,49) = 437.31$, $p < .001$ for AlexNet, $F(1,9) = 1139.43$, $p < .001$ for MobileNetV2, $F(1,9) = 222.62$, $p < .001$ for ResNet50). Further examining the difference between functionality measures, we see that the rank correlation values of decoding accuracy deficit are higher than that of network performance deficits ($F(1,49) = 79.22$, $p < .001$ for AlexNet, $F(1,9) = 669.25$, $p < .001$ for MobileNetV2, and $F(1,9) = 31.60$, $p < .001$ for ResNet50). Note that all CCA-based metrics are regularized in our formulations.

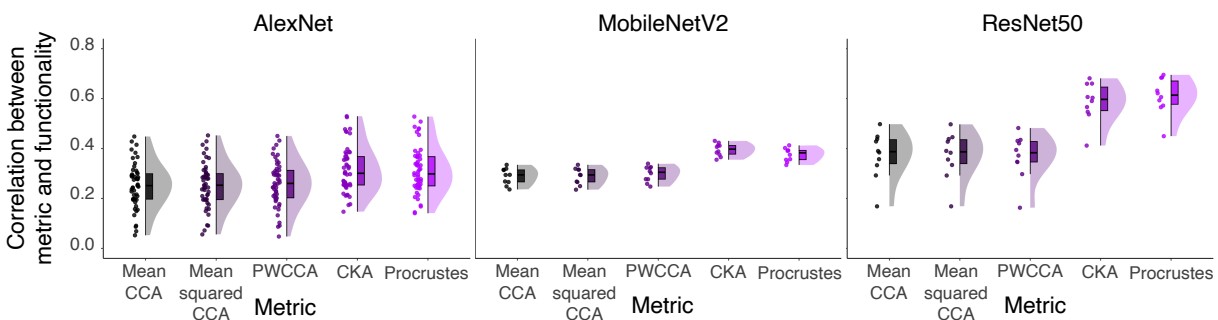

Figure 3: **CKA and Procrustes outperform CCA-based metrics.** This figure shows the distribution of rank correlation values across each metric, averaged between functionality measures on AlexNet (a), MobileNetV2 (b), and ResNet50 (c). Within each distribution, each data point represents the average rank correlation value from the ten classes, collapsed across network performance deficits and decoding accuracy deficits. Higher rank correlation values in the CKA and Procrustes condition indicate that these metrics are sensitive to the perturbations in the representations compared to other CCA-based metrics ($t(3392) = 9.00$, $p < .001$ for AlexNet; $t(10132) = 13.38$, $p < .001$ for MobileNetV2, $t(1572) = 25.41$, $p < .001$ for ResNet50). Note that all CCA-based metrics are regularized in our formulations.

Previously, Ding et al. (2021) performed some general stress tests on representation similarity metrics. In their work they showed that PWCCA distances were easily influenced by changes to random seed, predicting that the same layers in two different networks were more similar to other distant layers in the network than to each other. In another test, the CKA metric suffered from a different problem where it failed to distinguish representations from their low rank counterparts. Davari et al. (2022) pointed out other weaknesses associated with CKA. Among other findings, they demonstrated that CKA judged random and fully trained representations to be highly similar, especially in early layers of a network. Both studies used functional and intuitive tests for evaluation.

Although our work utilizes a framework introduced by Ding et al. (2021), we aim to test a fundamentally different hypothesis not addressed by their tests. Specifically, we aim to test how sensitive representation similarity metrics are to the functional properties of a representation actually used by a trained network during inference. In our tests we directly compare linear probe accuracy changes on in-distribution inputs to network performance changes on the same inputs after ablation. On the other hand, Ding et al. (2021) do not test this direct comparison. Instead, they perform more general stress tests on similarity metrics using either in-distribution linear probes or out-of-distribution network performance scores and different methods of perturbing representations. Through this direct comparison, we are able to show that similarity metrics are significantly less sensitive to the features of representations that are actually used by the network compared to its decodable features.

All of the RSMs we tested are significantly less sensitive to network performance changes induced by ablation than changes in class separability induced by ablation. This finding reflects the considerations made in developing these similarity metrics. RSMs were designed to compare the linear geometric properties of two representation spaces. So, it is not surprising that similarity metrics correlate with changes in decoding accuracies. On the other hand, network performance measures of function reflect how the network utilizes representations. In this case, functionally similar representations are those representations that remain similar after a series of non-linear transformations through layers of the network. Perhaps it is not surprising that this non-linear notion of similarity is harder to capture using current similarity metrics. However, it is the ultimate goal of interpretability to link representation and non-linear network function.

Previously, Bansal et al. (2021) proposed model stitching as a method for measuring the difference between two representations. Model stitching works by freezing two models and "stitching" the bottom half of one to the top half of another. Using this procedure the difference between the representation learned by the first model and the representation learned by the second at a particular layer is the drop in performance observed

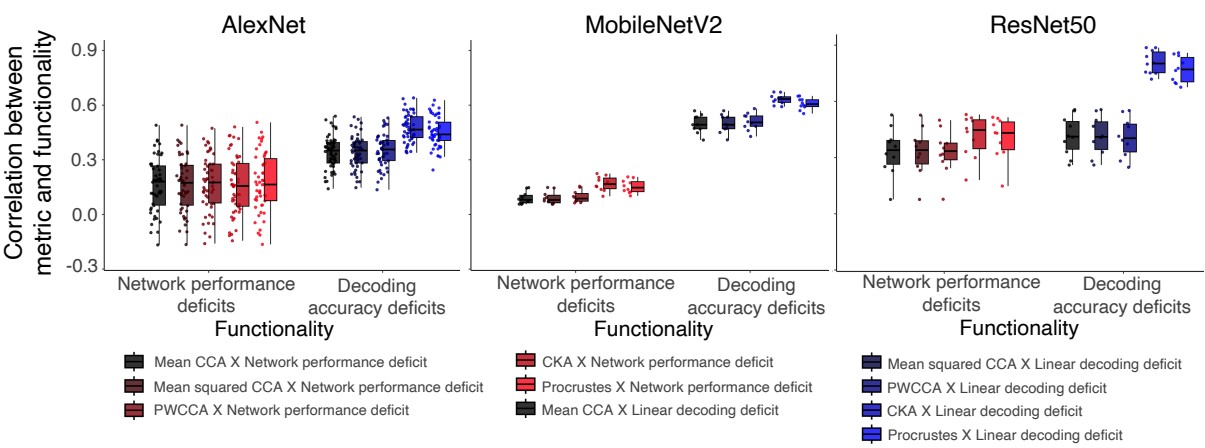

Figure 4: **Pattern of similarity metrics differs depending on functionality measure.** This figure shows a significant interaction between functionality and metric for Alexnet (top; $F(4, 3392) = 23.67$, $p < .001.$), for MobileNetV2 (middle; $F(4, 10132) = 4.59$, $p < .005.$), and for ResNet50 (bottom; $F(4, 1572) = 98.69$, $p < .001$). In the case of AlexNet, each of the five metrics behave differently depending on functionality type. The five similarity metrics within the network performance deficits condition are not significantly different, whereas metrics do differ within the decoding accuracy deficits condition. Specifically, the CKA and Procrustes metric is significantly different from the CCA-based metrics, only in the decoding accuracy deficit condition ($t(3392) = 13.10$, $p < .0001$). In both MobileNetV2 and ResNet50 models, there is also a significant interaction effect. Notably, CKA and Procrustes consistently outperform CCA-based measures, with a pronounced difference in decoding-based tests compared to network-based ones. Specifically, in decoding-based tests, the average correlation differences are $\beta = 0.16$ for MobileNetV2 and $\beta = 0.65$ for ResNet50 for CKA/Procrustes vs. CCA-based measures. In contrast, for network-based tests, these differences are smaller, with $\beta = 0.0865$ for MobileNetV2 and $\beta = 0.0817$ for ResNet50 with a diminished advantage for CKA and Procrustes based on network performance deficits compared to decoding accuracy deficits. However, CKA and Procrustes still significantly outperform CCA-based metrics on tests of network performance ($t(10132) = 6.534$, $p < .0001$ for MobileNetV2 and $t(1572) = 4.01$, $p < .0005$ for ResNet50). Note that all CCA-based metrics are regularized in our formulations.

for the stitched model compared to the original models. However issues arise when naively plugging one network into another. For instance, knowing which neurons in one network correspond to which neurons in another network is a non-trivial problem. This is especially true in cases where the two network architectures differ. These problems can be overcome by retraining some portion of the downstream network, as is done in Model Stitching. However, retraining brings with it the danger that the downstream function will change given the new representation. This can cause the decoding network to hallucinate features that were not used by the upstream network.

The problem of overly powerful decoders is especially pertinent in neuroscience where decoding studies have been performed for many decades. Neuroscientists take extreme caution to limit the power of their decoders to dissentangle functions performed by the brain from those hallucinated by the decoder (Tong & Pratte (2012)). By limiting the power of decoders, neuroscientists take the brain-centric perspective, restricting their search for brain function to computations the brain actually performs not those it could perform under different circumstances. We take a similar approach in this study by not retraining the downstream network on the ablated representation to reveal what information is actually used by the network and test RSMs to predict when this information is removed. Instead we find that RSMs are more attuned to changes in class separability revealed by linear decoders, the kind of computation that could be hallucinated by overly powerful decoders, but not used by the network.

Like previous studies, we show that some metrics tend to outperform others. Ding et al. (2021) previously showed that Procrustes tends to outperform CKA and CCA-based methods on language models. In our

experiments on larger image classification models, both CKA and Procrustes tend to perform better than regularized CCA-based methods adapted for larger representations. These results are hinted at by Ding et al. (2021) who show that CKA and Procrustes perform well on image decoding tests, but omit comparisons to CCA-based metrics.

Interestingly, the improved ability of CKA and Procrustes to capture function shrinks when tests of network function are applied. This reduced advantage occurs in every tested network and in AlexNet the advantage is completely eliminated. For AlexNet, network performance tests show that all the similarity metrics equally capture network function (Figure 4). This suggests that the advantages of CKA and Procrustes may be be overstated in previous literature. Furthermore, the best representation similarity metric may depend on the network used. To employ representation similarity metrics for interpretability, metrics should be developed that can capture the functional properties of representations across many networks. These tests can help future interpretability studies identify similarity metrics that can be used to infer the functional similarity of two networks based on their representational similarities.

## 5 Limitations

We acknowledge the following limitations in this work. Reformulating CCA-based measures to accommodate representations with more neurons than examples required using a regularized version of CCA called "ridge" CCA. In utilizing "ridge" CCA we had to choose regularization penalties to apply to each representation matrix. These penalties were chosen to be consistent across all tests, but were not cross-validated. Future works would benefit from testing more penalty settings to explore their effect on similarity results. In future work, we would like to explore these hyperparameter settings to establish best practices for computing RSM scores on large layers. Additionally, the current study focuses exclusively on investigating convolutional neural networks trained on ImageNet. Additional experiments should be conducted to determine how well the current findings generalize to new tasks and networks.

## 6 Conclusion

Taken altogether, the results of this study suggest that representation similarity metrics may already serve well as tools for comparing the geometries of representational spaces. However, RSMs are powerful tools that we would like to use to infer the functional similarity of two networks from their representational similarity. In other words, we would like to make the claim that two vision models "see" the same way if they have similar representations. In this work, we test RSMs using ablations on representations to determine if similar representations in CNNs are actually used the same way by downstream networks. Similarity metrics achieve high correlations with linear probe decoding accuracy changes in a representation induced by ablation. This sensitivity reveals that existing similarity metrics do a good job of predicting how useful two representations will be for linear downstream tasks. However, trained networks do not necessarily use the features of a representation that are relevant for linear decoding during inference (Hayne et al. (2022); Zhou et al. (2018); Meyes et al. (2020); Donnelly & Roegiest (2019)). This discrepancy reveals that representation similarity metrics could be improved by taking into account the features of representations used during inference.

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
