# OpenReview forum: "Does Representation Similarity Capture Function Similarity?"
_TMLR — Accepted by TMLR_

### Review · Reviewer_DRWf · 2023-10-28

**Summary Of Contributions:**

This work investigates what aspects are captured by representation similarity metrics (RSMs) of the form $\textrm{RSM}(A, B)$ where $A
\in\mathbb R^{n\times p_1}$ is a $p_1$-dimensional activation matrix of $n$ samples, and $B
\in\mathbb R^{n\times p_2}$ is a $p_2$-dimensional activation matrix of the same $n$ samples.

Specifically, the work seeks to understand to what extent RSMs capture the properties of:

* *Network performance* (i.e. what aspect of a given representation contributes to network performance), which is a measure of the “useful information” in the representation, and
* *Decoding accuracy* (i.e. what information about a given class is present in a representation in a linear dense), which is a measure of the “information present, independent of whether the network uses it” in the representation.

The work formulates this question in a statistical way:

* To what extent is *network performance* (rank) correlated with an RSM,
* To what extent is *decoding accuracy* (rank) correlated with an RSM, and
* Is the (rank) correlation of network performance with RSM higher than that of decoding accuracy with RSM.

In order to generate different representations to base the statistical analysis on, the authors perform interventions on the representation place, taking $A\rightarrow M\circ A=\widehat A$ where M is an element-wise masking matrix. This procedure can be performed many times. For any given mask $M$ instantiation, one can measure $RSM(A, M\circ A)$ as well as network performance deficit as $\textrm{Acc}(A) - \textrm{Acc}(M\circ A))$, which provides one sample for a correlation analysis. In summary, the authors are looking for statistical relationships between data generated by a causal mechanism of their design.

The authors investigate this for a number of different RSMs:

* Mean Canonical Correlation Analysis (CCA)
* Mean Squared CCA
* Projection Weighted CCA (PWCCA)
* Centered Kernel Alignment (CKA)
* Procrustes

In order to compute some of these RSMs, the authors introduce reduced complexity versions.

The authors find that:

* Similarity measures in general correspond more closely to linear decidability (i.e. information presence), than model performance (i.e. used information)
* CKA and Procrustes better correspond to aggregate model functionality

**Audience:**

Yes

**Broader Impact Concerns:**

I have no broader impact concerns.

**Claims And Evidence:**

Yes

**Requested Changes:**

# Required (critical)

* Perform a casual linear decoding experiment, to remove the possibility of the decoder and task modifications as a confounder to statements being made regarding causal versus non-causal. For a given layer, train a linear decoder on $X$. Now use the masking strategy to produce the $A$, and evaluate the linear decoder on these $A$ in the same way as you do in the non-causal setup through $\textrm{Dec}(X)-\textrm{Dec}(A)$.
* Explain the importance of a causal vs non-causal test (potentially use a causal DAG), and clearly distinguish this with a test of model downstream performance versus linear decoding.
* Expand the paragraph under “3 Results” to describe the method proposed in detail, to the level which the results can be reproduced. Please include:
    * What a hierarchical linear model is (with references)
    * How it is being used in your method (preferably presented alongside any relevant equations/cross-references)
    * Explain what exactly the Fisher Z transformation is being applied to.
    * Provide equations for the averaging process.
    * Explain what it means to “model classes as random units”.
* Explain all architectures and hyperparameters choices (see weaknesses section), and dependence (or insensitivity) of the drawn conclusions to these choices.
* Include a discussion on Model Stitching (see weaknesses section).
* Clarify in 2.2 which are contributions of this paper, and which are cited pre-existing contributions.
* Define all symbols introduced, including those of Section 3 which are statistics standards like $F$ and $t$. Explain what these symbols mean and how they relate to the analysis.

# Recommended for strengthening but not critical

* In the same spirit as the casual linear decoding experiment, perform a non-causal network performance experiment, and report $\textrm{Perf}(X) - \textrm{Perf}(A)$. In addition to the causal linear above, this would complete the question regarding whether the issue is the RSMs are having difficulty capturing causal measures or the semantics of the information. The non-causal network performance experiment may be prohibitively compute intensive, however, as one would need to (for example) fine-tune $f_{\textrm{Perf}}$ given a sampled masking $M$ for each $M$, and so this may be out of scope. Fine-tuning hyperparameter choices may also introduce additional confounders in this setting.
* Evaluate the Model Stitching method using the testing protocol.
* Page 1, 3rd line of 2nd paragraph. “... invariant under linear transformations while others ...” → “... invariant under invertible linear transformations while others ...”
* Page 3, Increase the resolution of Figure 1, or use vector graphics.
* Page 4, point 1. re-add the space $\mathbf X$ lives in.
* Page 4, point 2. Define $A$. Is $A$ a set? If so, use set notation, e.g. $\mathbb A$.
* Page 4, point 3.1, for $D$ to be defined as given, it needs an $i$ subscript to correspond to a given $A_i$.
* Page 4, point 3.2, $F_{Perf}$ →  $F_{Perf}(\mathbf X, \mathbf A_i)$ (it is not a global quantity) and add subscript i inside mod.
* Page 4, point 3.3, $F_{Dec}$ →  $F_{Dec}(\mathbf X, \mathbf A_i)$ (it is not a global quantity) and add subscript i inside mod.
* Page 4, point 3.4, be clear this is Spearman rank which you use later (and do not have a linear relationship prior).
* Page 5, Equation (6) $\rho_i$ → $\rho_i(\mathbf A, \mathbf B)$, this equation can go on a single line.
* Page 7, Increase the resolution of Figure 2, or use vector graphics. Increase the size of the font for the y-ticks. You can use a shared y label here to give you additional horizontal space. Consider tying the y-axes in order to make the different models comparable.
* Page 8, Increase the resolution of Figure 3, or use vector graphics. Increase the size of the font for the y-ticks. You can use a shared y label here to give you additional horizontal space. Consider tying the y-axes in order to make the different models comparable. Alternatively, the ResNet y-axis can be rescaled as the data currently occupies only a fraction of the plot.
* Page 9, Increase the resolution of Figure 4, or use vector graphics. I found it challenging to understand the information being conveyed in this plot. I can see the statistical difference between CKA + procrustes and the other methods for network performance in the case of non-AlexNet architectures, however, this many cross-references against legend and color matching. It would be helpful for the reader if this plot could be more simply presented without information loss.
* Throughout, be careful about tildes and indices, for example top of page 6. $\tilde{\alpha_i}$ → $\tilde{\alpha}_i$.
* Move statistical results to figure captions, and reserve main prose for discussion of the results.
* Check for different architecture types within problem. E.g. perform the same analysis with a non-conv architecture, for example a PyTorch ViTB-16/224 ImageNet pre-trained model can be found in the time library https://timm.fast.ai/create_model.
* Check a different problem setting. E.g. perform the same analysis in a non-image setting, for example, using a pre-trained BERT model from HuggingFace on a small-medium sized text corpus.
* Give clear, numbered, reference-able (and then reference) definitions for:
    * Causal test
    * Non-causal test
    * Functionality
    * Network Performance
    * Linear Recoding
    * RSM
* Provide discussion around why AlexNet behaves qualitatively differently to MobileNetV2 and ResNet50.
* Give guidance on how researchers going forward should use RSMs based on your conclusions. Discuss what quantitative conditions are necessary for an RSM to meet in order for an RSM to have a given utility.

**Strengths And Weaknesses:**

# Strengths

The work sets out to measure quantities of broad interest to the community: What are different RSMs capturing and not capturing? The work is set out as an empirical study, with an experimental protocol of how quantities of interest will be measured and aggregated up.

The work contains a nice discussion and summary of some widely-used RSMs, some related numerical approximations, and results on what aspects of the data those RSMs are capturing.

# Weaknesses

The work focuses on evaluating sensitivity to causal function changes. However, the work only investigates the causal form of the network performance functionality, whilst opting for a non-causal test for linear decoding functionality. One of the core conclusions of the paper is that the RSMs are not capturing the causal non-linear network functions. Yet, because in their analysis the linear decoding benchmark is both non-causal and linear, it is not possible to say whether the ineffectiveness of the RSM is due to the network non-linearly, or due to a swap from causal to non-causal tests. This test chosen testing protocol obfuscates any general conclusions that can be drawn.

Independent of evaluation, the general importance of causal versus non-causal tests of representations is not discussed independently to the importance of capturing network performance.

Additionally, the tasks are different across these choices (full classification, and one-versus rest classification respectively), potentially introducing further confounding into any overall statements regarding precisely which aspects RSMs are failing to capture.

I am happy to see the acknowledgement of the limitations of the CCA approximation used. Using the approximation itself is not a weakness, however, the work does not contain any discussion regarding the extent to which the approximation is valid, and in the scenarios being tested, the size of approximation errors that should be expected (and crucially, whether these approximation errors could dominate the statistical analysis, either in terms of systematic bias or noise).

Reasons are not given for the specific experiments performed (linear probe vs non-linear probe), architectures chosen (why AlexNet, MobileNetV2, ResNet50), hyperparameters chosen (e.g. 50 classes chosen for AlexNet compared to 10- classes for MobileNetV2 and ResNet50). Are the results sensitive to these hyperparameters? Does the number of samples chosen, features chosen (100, randomly sampled - each time or fixed?), or repetitions (200) meet some numerical convergence criteria? The paper does not contain the details that convince a reader these choices are natural or that the key conclusions do not depend on their choice.

The conclusion contains “representation similarity metrics may already serve well as tools for comparing the geometries of representational spaces, but could be enhanced in order to capture causal, non-linear network function.”.
The model comparison Model Stitching (https://arxiv.org/abs/2106.07682) was introduced for this reason. A discussion of how Model Stitching would fare under the process outlined in Figure 1, or better, an evaluation of this representation comparison methodology feels missing.

Analysis is performed on a single dataset (ImageNet) with three similar (conv-based) architectures, yet general conclusions are being drawn.

It is unclear what a reader should do given the conclusions (should we stop using RSMs/should we develop others/does an RSM exist than can meet the requirements of the test?). To what extent is a non-correlation a problem (what numbers do we need for the RSM to be useful in practice?).

The paragraph under “3 Results” is difficult for me to follow and contains key steps required for reproducing the remainder of the section, which obfuscate details required to meet reproducibility standards.

Key experimental quantities leading to the conclusions drawn are buried within paragraphs, and accompanied with a number of distracting undefined (in text) quantities. The experimental discussion would be easier to follow if statistical quantities were associated with figure captions, and the prose mostly reserved for discussions of the results, highlighting the main points.

The claim around Equation (3) as their derivation enabling computation of CKA on larger layers than prior works is confusing to me, as the original CKA paper (https://arxiv.org/abs/1905.00414) contains this property as the second equality in Equation (1). Similarly, the way the kernel trick for calculating Equation (6) as presented in the paper presents the contribution as one of the current work, however, enabling CCA in high dimensions is the work of the (cited) https://arxiv.org/abs/2011.01650. This should be clearer.

The writing could be made more succinct, with the important concepts pulled out, numbered, labeled, and cross-referenced. For example, the definitions of Linear Decoding Functionality Measures, and Network Performance measures are mentioned (but not linked, numbered, or referable to in an unambiguous way).

---

> ### Author Response · Authors · 2023-11-02
> **Thank you and clarification on requested changes**
>
> Hello DRWf,
>
> Thank you very much for your thoughtful and detailed review. We are in the process of integrating your requested changes into a revised manuscript. After other reviewers have submitted their reviews, we will upload a revised manuscript as well as detailed comments addressing your concerns.
>
> As we integrate your requested changes into the manuscript, we would appreciate your guidance on the following points.
>
> 1. Could you clarify what is meant by your first requested change: "perform a casual linear decoding experiment." In your current explanation, you suggest that a causal linear decoding experiment can be conducted by training a linear decoder on matrix $X$ ($\text{Dec}(X)$), training a linear decoder on matrix $A$ ($\text{Dec}(A)$), and comparing the results ($\text{Dec}(X)-\text{Dec}(A)$). This is the same process we used for our test of linear decoding functionality in the current manuscript (See Figure 1). What specific changes would you recommend making to differentiate this new test from our current linear decoding test?
>
> 2. Can you elaborate on what kind of causal DAG you would like to see to support the importance of causal tests? Do you have an example in mind? It is not clear to us how one would formulate our problem as a DAG. In our work we are testing two independent relationships. First, we test whether the RSM difference between two representations can explain linear decoding differences between the same representations. Second, we test whether the RSM difference can explain the downstream network performance differences between the two representations. We do not assume in our analysis that the linear decoding difference mediates the relationship between the RSM difference and the network performance difference. Instead, we test these relationships independently and measure the total effect of RSM differences in each case.
>
> Thank you very much for your time. We apologize for needing the extra explanation but want to make sure we properly address your concerns.

---

> ### Comment · Reviewer_DRWf · 2023-11-06
> **Response to request for clarifications**
>
> Thanks for following-up/clarifying. Please find below clarifications for the your points. Please let me know if anything is still unclear.
>
> ## Clarification 1
>
> If I understand correctly, in your paper you perform:
>
> 1. A causal measure of function through $F_{Perf}$, and
> 2. A non-causal measure of decoding through $F_{Dec}$.
>
> The core difference between the two measures from an evaluation perspective is how the masking neurons are used. Respectively:
>
> 1. $F_{Perf}$ compares the performance of applying the model remainder to masked inputs compared to unmasked inputs. The model remainder *is not trained on the masked inputs* $A_i$ (Section 2.4, second paragraph).
> 2. $F_{Dec}$ compares the linear decoding performance on masked inputs compared to unmasked inputs. The linear decoder *is trained on the masked inputs* $A_i$ (Section 2.4, first paragraph).
>
> To the best of my understanding, whether a test is causal or not depends on if whether model updates are exposed to the masking strategy or not:
>
> * Not exposing the model to the masking strategy, and intervening on the inputs implies a causal test.
> * Exposing the model to the masking strategy, and evaluating at that mask implies a non-causal test.
>
> Please let me know if I have misunderstood the above.
>
> Assuming that the above is correct, a causal measure of linear decodability would train a linear decoder on the unmasked inputs, and evaluate the linear decoder on the masked inputs $A_i$. If we denote $\text{Dec}(X_1; \theta(X_2))$ as the linear decode applied to $X_1$ given parameters $\theta(X_2)$ determined by $X_2$, then a causal test would look like
>
> $$|\text{Dec}(X; \theta(X))-\text{Dec}(A;\theta(X))|$$
> whereas your Figure 1 instead computes
> $$|\text{Dec}(X; \theta(X))-\text{Dec}(A;\theta(A))|$$
> which is a *non-causal* test of linear decodability.
>
> If my understanding above is incorrect, are you able to provide a definition of what makes a test *causal* and *non-causal* in the context of your work?
>
> ## Clarification 2
>
> The discussion of “causal tests” implies consideration of Peters et al. *Elements of Causal Inference* [1] or Pearl et al. *Causal Inference in Statistics* [2]. In each of these cases, a causal effect of $X$ upon $Y$ is formulated in the do calculus through $P(Y | \textrm{do}(X = \textrm{True}))$ and contrasting this to the observed $P(Y | X=\textrm{True})$. The do represents an intervention, and any causal link verified in this way is encodable in a DAG where the arrows represent non-zero instances of $P(Y | \textrm{do}(X = \textrm{True}))$ with respect to given random variables $X$ and $Y$.
>
> In your setting, I think you have the following nodes:
>
> * The input $X$
> * The model performance $A$
> * The linear decoding performance $B$
>
> with the DAG structure $X \rightarrow A$, $X \rightarrow B$.
>
> Asking if intervening on X with a mask affects mode performance can be formulated as evaluating the quantity $Q=|P(A | X) - P(A|\textrm{do}(X=A)) \geq \gamma$ for some threshold $\gamma$ and mask $A$. Alternatively, one can just report values of $Q$, and then (as you do) compute the correlation of $Q$ with related quantities regarding linear decoding performance.
>
> One other clarification is that my main suggestion here is to represent the specific causal test you propose in the natural language of causality of Pearl and others, which is causal DAGs and the do-calculus, as opposed to putting the entire correlation analysis of the paper into a causal DAG.
>
> [1] https://mitpress.mit.edu/9780262037310/elements-of-causal-inference/#:~:text=Elements%20of%20Causal%20Inference%20is,data%20to%20understand%20the%20world
>
> [2] http://bayes.cs.ucla.edu/PRIMER/

---

> ### Author Response · Authors · 2024-01-24
> **Revised Manuscript Upload (Part 1)**
>
> Thank you very much for your thoughtful, detailed, and constructive feedback. We hope to address your concerns in three categories based on which aspect of the paper could be improved: motivation, clarity, and experimentation. We will address each category of concerns separately with an explanation of the associated changes we have made to the manuscript. At the end of this response we detail the miscellaneous changes made to the manuscript in response to your feedback.
>
> # Motivation
>
> First, we would like to improve the motivation for our work in the revised manuscript by addressing some of your concerns. We start by clarifying our general motivation based on your feedback and finish with how this motivation inspires the experiments we run. In the weaknesses section of the review, you write:
>
> >**_Independent of evaluation, the general importance of causal versus non-causal tests of representations is not discussed independently to the importance of capturing network performance._**
>
> To help motivate the "importance" of causal and non-causal tests of RSMs, we have added the following text to the introduction section of the updated manuscript:
>
> > _Representation similarity metrics (RSMs) have been used to help answer questions like “Do vision transformers see like convolutional neural networks” (Raghu et al. (2021)). Put another way, researchers aim to answer whether two different networks function in a similar way. Two networks can be shown to function in a similar way when two criteria are met. First, the two networks compute similar representations. What it means for two representations to be similar, however, is not straightforward. In response to this vagueness, many RSMs have been developed with different notions of similarity over the past decade. For example, some RSMs are invariant under invertible linear transformations while others are not (see Kornblith et al. (2019) for a theoretical comparison). Capturing the various notions of representational similarity has been the primary driver behind developing and testing RSMs._
> >
> > _Second, to characterize two networks as functioning similarly, an additional criterion must be met. The two networks must also use their representations in a similar way. This second criterion has been mostly neglected when considering the similarity of network function using RSMs. In this paper, we aim to investigate whether RSMs can tell us when this second criterion is met. In other words, how likely is it that two representations deemed similar by RSMs are actually used the same way by trained networks?_
>
> We hope that this revised text illustrates the motivation for our approach. To quantify the likelihood that two representations deemed similar by RSMs are actually used the same way by trained networks, we perform interventions (zero ablations) on neurons in each layer of trained CNNs. As you pointed out (**Required (critical) #2**), these interventions should be represented using Pearl's do-calculus (Pearl (2009)). Accordingly, we have updated Section 2.1 of the manuscript with the following text:
>
> >_Interventions on $X$ come in the form of zero ablations where entire features (columns of $X$) are set to zero using a mask producing an ablated representation $A$ where $A = M \otimes X$. The functionality of the masked representation is evaluated using the downstream neural network. We represent the zero ablation intervention on functionality with the do$(\cdot)$ operator. For each image $x$, we compute the predicted target class rank $r_t(x)$ for the target class $t$. For instance, if the CNN's softmax layer predicts that the Junco bird class was the third most likely class given an image of a Junco bird, then that image receives a rank of three ($r_t(x) = 3$). The impact of the ablation on function is registered as the difference in predicted target class ranks produced by the intervention on the representation._
> >
> > $$ \Delta_\text{ablate} = [ r_t(x) - r_t(x | \text{do}(A)) ] $$

---

> ### Author Response · Authors · 2024-01-24
> **Revised Manuscript Upload (Part 2)**
>
> We hope that this information helps clarify the motivation and notation for ablation interventions on the network. In the previous manuscript, we referred to these interventions as causal tests. However, after considering your feedback, we would like to transition away from the use of the word "causal" in this context. Interventions like those employed in this study are typically used to infer the causal role of representations (McGrath et al. (2023)). Here, we do not aim to test the causal role of representations directly. Instead, we use interventions on representations to test RSMs for their ability to capture the second criterion mentioned previously for inferring functional similarity from representational similarity. In the context of the current study, our interventions are not causal because they are not aimed at revealing causal properties of representations. For this reason, we omit the term "causal" in our revised manuscript.
>
> Similarly, we avoid the use of the term "non-causal" in the revised manuscript. In our original manuscript, non-causal tests were those that involved processing representations using linear decoders rather than the downstream network. We originally called them "non-causal" tests because interventions that affect linear decoders do not necessarily "cause" network performance changes. After reflecting on the terminology, we have concluded that the term "non-causal" is inappropriate because we do not study causality specifically using these tests. Instead, we use linear probes as a measure of class separability in the representation. We have added the following text to the introduction to help motivate our use of linear decoders for testing RSMs.
>
> > _In this work, we aim to quantify how likely is it that two representations deemed similar by RSMs are actually used the same way by trained networks. Previous studies have tackled this question by making assumptions about what kinds of information networks use. If this information is detectable using RSMs, then RSMs can be used to determine if two representations are used by the network in the same way. For instance, RSMs have been evaluated for their ability to track class separability in representations (Ding et al. (2021)). To measure the class separability in a representation, researchers usually train linear probes for downstream tasks on learned representations and compare the results. However, the features of a representation that carry the most information for linear probes may not be those actually used by the network during inference. Studies that remove features from representations in trained networks, called ablation (LeCun et al. (1989)), have revealed a weak link between the relevance of a feature for decoding and its effect when removed from the network (Meyes et al. (2020); Zhou et al. (2018); Donnelly & Roegiest (2019); Morcos et al. (2018b)). Hayne et al. (2022) recently showed that linear decoders specifically cannot single out the features of representations actually used by the network. Consequently, two representations that are equally decodable using linear probes may not actually be equal from the point of view of network performance._
>
> > _If networks neglect linearly decodable information, then RSMs that track linear class separability may not satisfy the second criterion mentioned previously. Remember the second criterion says that to infer the functional similarity of two networks from RSM scores, one must also show that networks use similar representations in similar ways. In this work, we propose using ablations to directly test RSMs to satisfy the second criterion. We use ablation to evaluate how closely changes in RSM scores are related to changes in network performance. We first ablate groups of units from either AlexNet, MobileNet, or ResNet50, compare the original representations to the ablated representations using RSMs, and then compare scores to the changes seen for network performance (see Figure 1). As a baseline, we also compare RSM scores to changes in class separability in the representation using linear probes. This process reveals the extent to which RSMs satisfy the second criterion, tracking functional similarities as opposed to just linear similarities between representations. This distinction is crucial for neural network interpretability where the aim is to develop human-understandable descriptions of how neural networks actually rely on their internal representations._

---

> ### Author Response · Authors · 2024-01-24
> **Revised Manuscript Upload (Part 3)**
>
> We hope that that this additional text helps motivate the way in which we deploy linear decoders. In your review you write:
>
> > **_The work only investigates the causal form of the network performance functionality, whilst opting for a non-causal test for linear decoding functionality._**
>
> The reason we retrain the linear decoders after the interventions is so that we can measure the effect of the ablation on class separability. The linear decoding test reveals that RSM scores correlate with class separability changes due to ablation. This is similar to what has been shown previously for RSMs (Ding et al. (2021)). However, this correlation does not imply that RSMs satisfy the second criterion required for inferring functional similarity from representational similarity of two networks.
>
> To address this question of whether functional similarity can be inferred from representational similarity, it would be ideal to plug one network into another to determine if the representation from one network is used by another. However, issues arise when naively plugging one network into another. For instance, knowing which neurons in one network correspond to which neurons in another network is a non-trivial problem. This is especially true in cases where the two network architectures differ. These problems can be overcome by retraining some portion of the downstream network, as is done in Model Stitching (Raghu et al. (2021)). However, retraining brings with it the danger that the downstream function will change given the new representation (we address this concern and others in the discussion section of our revised manuscript (**Required (critical) #5**)).
>
> The problem of overly powerful decoders is especially pertinent in neuroscience where decoding studies have been performed for many decades. Neuroscientists take extreme caution to limit the power of their decoders to disentangle computations performed by the brain from those hallucinated by the decoder (Tong and Pratte (2011)). By limiting the power of decoders, neuroscientists take the brain-centric perspective, restricting their search for brain function to computations the brain actually performs not those it _could_ perform under different circumstances. We take a similar approach in this study by not retraining the downstream network on the ablated representation to reveal what information is actually used by the network and showing that RSMs do predict when this information is removed. In contrast, RSMs are more attuned to changes in class separability revealed by linear decoders, the kind of computation that could be hallucinated by overly powerful decoders, but not used by the network.
>
> With all this in mind, we acknowledge your request for a linear decoding experiment in which the linear decoder is not retrained (**Required (critical) #2**). This could help reveal exactly why RSMs fail the ablation experiments we perform. We have been conducting this experiment since your initial review. We hope to include these results in a future publication but are unable to finish the analysis in time for the revised manuscript (see the Experiments section of this response).

---

> ### Author Response · Authors · 2024-01-24
> **Revised Manuscript Upload (Part 4)**
>
> # Clarity
>
> Second, we would like to improve the clarity of our revised manuscript by addressing some of your concerns. In your review, you write:
>
> > **_Expand the paragraph under “3 Results” to describe the method proposed in detail, to the level which the results can be reproduced._**
>
> In the updated manuscript we have added text to clarify the specific points you raised (**_in bold_**):
>
> > **_Hierarchical model and factors_**
> In this section, we detail the results of aggregating our statistical tests across each layer and class from AlexNet, MobileNetV2, and ResNet50. Our approach was centered around a hierarchical linear model to evaluate the prediction of the rank correlation values as a function of two primary factors: Metric and Functionality. We will refer to the term “factor” to illustrate metric and functionality, whereas we will refer to the term “level” to illustrate the number of embedded conditions within each factor.
>
> > **_Factor description_**
> Metric: refers to the computations utilized to establish representational similarity between two representations. In our study, the metrics refer to five different measures of CKA, Procrustes, and regularized CCA-based methods of mean CCA, mean squared CCA, and PWCCA.
>
> > **_Factor description_**
> Functionality: refers to the way one can test a representation as opposed to ablated representations. In our study, we have two conditions: first, Network performance, which is comparing the performance of a network with intact representations versus the performance of a network with ablated representations fed back into that very network.  The second level of the functionality is Linear decoding, where a linear probe is fit to the intact and ablated representations.
>
> > In total, we have a factorial experimental design with 2 functionality (network performance/linear decoding) X 5 metrics (CKA/Procrustes/mean-CCA/mean-squared-CCA/PWCCA). This factorial design and analysis allows us to test the average effects of functionality, metric, and the interaction of these factors and pinpoint the effects of these factors on the correlation between representational similarity and network performance change or linear decoding change with intact or ablated representations.
>
> > **_transforming dependent variable_**
> To ensure the comparability of correlation coefficients, we fisher-z transformed each rank correlation value. This transformation normalizes the distribution of coefficients, making them suitable for averaging and further statistical analysis.
>
> > **_summary statistics_**
> >Once Fisher-z transformed, we implement a hierarchical linear model as a function of functionality and metric, while allowing for random intercepts and random slope of functionality.
>
> >In which:
>
> >$$FisherZ_{ij} = \beta_0 + \beta_1 \times Functionality_{ij} + \beta_2 \times Metric_{ij} + \beta_3 \times (Functionality_{ij} \times Metric_{ij})  + u_{0j} + u_{1j} \times Functionality_{ij} + R_{ij}$$
>
> See the revised manuscript for full details.

---

> ### Author Response · Authors · 2024-01-24
> **Revised Manuscript Upload (Part 5)**
>
> > **_cluster description_**
> In our model, classes were treated as clusters, which acknowledges that variability also exists within each cluster as opposed to between clusters. With this approach, allocating the correct variance to the factors of interest and the to within subject variance is possible, allowing for a robust analysis.
>
> > **_transforming dependent variable_**
> The transformed correlation values were modeled as a function of metric type and functionality measure. This modeling helps us understand whether the correlation between  representation similarity and functionality differs as a function of using different metrics to calculate representational similarity. In summary, this approach allows us to not only compare different metrics and functionality measures but also to pinpoint those that are most indicative of significant changes in neural representations. Such insights are invaluable in advancing our comprehension of neural network architectures and their practical applications.
>
> > **_package description_**
> All analyses were conducted using the “lme4” package in R (Bates et al. (2014)). Figures were generated using “ggplot2” (Wickham (2016), “raincloudplots” (Allen et al. (2021)), and “smplot” package (Min & Zhou (2021)) in R.
>
> Next, we address comments aimed at clarifying hyperparameter choices (**_Required (critical) #4_**). Below we justify specific hyperparameter choices and note that corresponding changes have been made in the updated manuscript.
>
> * We chose our three networks as a representative sample of three convolutional networks with different architectures that are still widely studied for producing, testing, and analyzing representation similarity metrics (Kornblith et al. (2019); Raghu et al. (2021); Boix-Adsera et al. (2022)).
> * We chose 50 classes for AlexNet and 10 classes for MobileNet and ResNet to strike a balance between convenience and breadth. Computing RSM scores requires working with a representation matrix $X \in \mathbb{R}^{m \times n}$, where $m$ is the number of images and $n$ is the number of neurons in a given layer. Given the large layer sizes of MobileNet and ResNet (~100,000 neurons), adding more classes increases the cost of computing RSM scores. To help avoid bias in our small sample of classes, we sample the classes completely randomly from ImageNet. We acknowledge that increasing the number of images is desirable and we plan on including more images in future work, however, we note that previous studies include a comparable number of images in their analysis (Raghu et al. (2021)).
> * Next, we address concerns regarding hyperparameters chosen for the linear decoding experiments. As mentioned previously the linear probe experiments are designed to estimate the extent to which layer features linearly separate the class of interest from other classes. To estimate this we chose to randomly sample neurons from each layer (as in Alain and Bengio (2016)) and fit a one-vs.-rest linear probe to the features (as in Dosovitskiy et al. (2020)). This process is computationally efficient. In contrast, a full classification linear probe fit to the entire layer is expensive to train for large layers and does not directly reveal the average linear separability of the target class for features in the layer.
> * We choose regularized versions of CCA to estimate representational similarities in our analysis for computational reasons outlined in the manuscript. There is some concern (**_Weaknesses #4_**) this estimation could introduce errors into our results that interfere with the conclusions of the paper. We have run additional experiments to test different penalties for regularizing the CCA metrics. We find that the exact CCA values do change when the penalties change. However, in the current study, we are not concerned with the exact CCA values, only the relative order of scores given an ablation. We find that CCA metrics regularized with different penalties are highly correlated (Spearman rank correlation > 0.95). In future work, we would like to expand on these results to establish best practices for computing RSM scores on large layers.
>
> # Experiments
>
> In this section, we address requests for additional experiments. First, we acknowledge the importance of running a causal linear decoding experiment (**Required (critical) #1**). As you mention, running a causal linear decoding experiment would help tease apart the factors that account for the differences between our two experiments. Since your initial review, we have run these experiments on a supercomputer at our institution. Unfortunately, training a full classification linear decoder on each layer of each network and computing RSM scores is computationally expensive. Accordingly, we recently generated preliminary results, but we don't yet trust the accuracy of the results enough to include them in the paper. We hope to include these results in a future publication devoted to isolating the effects of ablations on RSMs.

---

> ### Author Response · Authors · 2024-01-24
> **Revised Manuscript Upload (Part 6)**
>
> In our current paradigm, we would still like to argue that a comparison between retrained linear decoders and frozen downstream networks produces valuable insights. The performance of the retrained linear decoder gives us a rough upper bound on the class-specific linear information remaining in the representation after ablation. Whereas the performance of the frozen downstream network tells us how much class-specific information remains that the network actually uses. Ideally, we'd like to be able to use RSMs to infer when two networks are functionally similar. If, however, RSM scores predict the presence of class-specific linear information better than the presence of class-specific information used by the network, then RSMs are less reliable when used to infer when two networks are functionally similar. Instead, they would be best used to infer when representations have similar class-specific linear information.
>
> In addition, we do not have enough time to run experiments in which the downstream network is retrained after every ablation (**_Recommended for strengthening but not critical #1_**) or Model Stitching experiments (**_Recommended for strengthening but not critical #2_**), although we have discussed Model Stitching in the revised manuscript. These are important experiments that we would like to include in future work.
>
> # Miscellaneous
>
> > **_Page 1, 3rd line of 2nd paragraph. “... invariant under linear transformations while others ...” → “... invariant under invertible linear transformations while others ...”_**
>
> Updated in revised manuscript.
>
> > **_Page 3, Increase the resolution of Figure 1, or use vector graphics._**
>
> We have replaced all the figures with vector graphics in the updated manuscript.
>
> > **_Page 4, point 1. re-add the space $\mathbf{X}$ lives in._**
>
> We're not exactly sure what you mean here. We're happy to make the change for the camera-ready paper.
>
> > **_Page 4, point 2. Define $A$. Is $A$ a set? If so, use set notation, e.g. $\mathbb{A}$._**
>
> Thank you for pointing this out. $A$ is indeed a set. Manuscript updated to clarify.
>
> > **_Page 4, point 3.1, for $D$ to be defined as given, it needs an $i$ subscript to correspond to a given $A_i$._**
>
> Updated in revised manuscript.
>
> > **_Page 4, point 3.2, $F$ → $F(X,A_i)$ (it is not a global quantity) and add subscript i inside mod._**
>
> Updated in revised manuscript.
>
> > **_Page 4, point 3.3, $F$ → $F(X,A_i)$ (it is not a global quantity) and add subscript i inside mod._**
>
> Updated in revised manuscript.
>
> > **_Page 4, point 3.4, be clear this is Spearman rank which you use later (and do not have a linear relationship prior)._**
>
> Updated in revised manuscript.
>
> > **_Page 5, Equation (6) $\rho_i$ → $\rho_i(\mathbf{A},\mathbf{B})$, this equation can go on a single line._**
>
> Updated in revised manuscript.
>
> > **_Page 7, Increase the resolution of Figure 2, or use vector graphics. Increase the size of the font for the y-ticks. You can use a shared y label here to give you additional horizontal space. Consider tying the y-axes in order to make the different models comparable._**
>
> We have replaced all the figures with vector graphics and made the requested changes in the updated manuscript.
>
> > **_Page 8, Increase the resolution of Figure 3, or use vector graphics. Increase the size of the font for the y-ticks. You can use a shared y label here to give you additional horizontal space. Consider tying the y-axes in order to make the different models comparable. Alternatively, the ResNet y-axis can be rescaled as the data currently occupies only a fraction of the plot._**
>
> We have replaced all the figures with vector graphics and made the requested changes in the updated manuscript.
>
> > **_Page 9, Increase the resolution of Figure 4, or use vector graphics. I found it challenging to understand the information being conveyed in this plot. I can see the statistical difference between CKA + procrustes and the other methods for network performance in the case of non-AlexNet architectures, however, this many cross-references against legend and color matching. It would be helpful for the reader if this plot could be more simply presented without information loss._**
>
> We have replaced all the figures with vector graphics in the updated manuscript. We considered different strategies for making this figure more readable. In the updated manuscript we have tied the y-axes, but are open to additional requested changes for the camera-ready paper.
>
> > **_Throughout, be careful about tildes and indices, for example top of page 6. $\tilde{\alpha_i}$ → $\tilde\alpha_i$._**
>
> Thank you for catching this. We have updated in the revised manuscript.
>
> > **_Move statistical results to figure captions, and reserve main prose for discussion of the results._**
>
> We have tried to accommodate this request in the updated manuscript. If there are changes we can make for the camera-ready paper, let us know.

---

> > ### Comment · Reviewer_DRWf · 2024-02-02
> > **Response to revisions**
> >
> > Thank you for working to update the manuscript.
> >
> > The rewording regarding causal versus non-causal to instead linear separability versus functional similarity has helped improve the overall clarity of the work.
> >
> > As this work is now not necessarily claiming something explicitly causal, I no longer think it is necessary to run the non-retrained linear decoder analysis that I originally requested, although I do think the results of this experiment would be very interesting to see in future work.
> >
> > Other revisions (expansions to the manuscript, providing reworked figures in vector graphics, moving many of the statistical results to the captions) have improved the overall clarity.
> >
> > Regarding the query about Page 4 point 1, in the current revision, the suggested change could be $X \rightarrow X\in \mathbb R^{n\times p_{\textrm{input}}}$ for the first bullet point now at the top of page 5. Similarly, adding the space that the activations $A$ live in could help.
> >
> > Other typos/wordings I spotted upon re-reading:
> >
> > * Penultimate paragraph bottom of page 2:  “we propose using ablations to directly test RSMs to satisfy” → “we propose using ablations to directly test if RSMs satisfy”
> > * Final paragraph page 4. I think you want your mask symbol to be $A=M\otimes Z$  → $A=M\odot Z$ as the former usually implies a combinatorial or cross-product, and this is usually considered element-wise (which is what I think you are doing)
> > * Bottom of page 7 “($\tilde100,000$ neurons)” → “(~$100,000$ neurons)”
> >
> >
> > One final thought is that given the work has mostly moved away from specific causal claims, is it worth considering a different title, for example: “Does representation similarity capture function similarity?”

---

> > > ### Author Response · Authors · 2024-02-02
> > > **Manuscript updated**
> > >
> > > We appreciate your thorough engagement with the revised manuscript. The title you proposed does a much better job capturing the aim of the paper. The title as well as your suggested revisions are reflected in the latest update to the manuscript. Please let us know if you notice anything else.

---

### Review · Reviewer_x8XN · 2023-12-20

**Summary Of Contributions:**

The paper presents an empirical comparison of representation similarity metrics
for deep neural networks. The authors draw conclusions with respect to the
sensitivity of the metrics, and how well they capture the underlying semantics.

**Audience:**

Yes

**Claims And Evidence:**

No

**Requested Changes:**

- Please explain in detail how what you are considering is causal.
- Please present the similarity values.
- Please clarify the details of the statistical testing.
- Please describe in detail how performance deficits were computed.
- Please evaluate the impact of normalization on the metrics.
- Please make the scales in Figure 4 consistent.

**Strengths And Weaknesses:**

The paper is well-written, although it was not clear to me how the changes the
authors are inducing in the networks are causal and to what extent the tests
that the authors use are causal. This deserves more discussion as it is the
central claim of the paper.

For the presentation of results, it would be great to see the actual similarity
values in addition to the correlation coefficients. In particular, it is unclear
whether the similarity values are what you would expect at the moment, i.e. more
similar networks have higher similarity scores. There could be a good
correlation even if the actual values do not make sense, so it is important to
show them.

The statistical tests the authors run are unclear. The authors say that they
compute the Spearman correlation (rank?) coefficient, but also report p-values
-- how were those computed? Did you correct for multiple testing?

Similarly, it was unclear to me how the performance deficits were produced.
Please elaborate.

What is the impact of regularization on the CCA-based metrics? The authors note
that they perform worse than CKA and Procrustes; could this be a consequence of
the normalization?

For Figure 4, please use the same scale for each sub-plot -- it looks like the
differences for MobileNetV2 are largest, but this is just an artifact of the
different scale.

---

> ### Author Response · Authors · 2024-01-17
> **Thank you and clarification on requested changes**
>
> Thank you very much for your thoughtful review and for recognizing strengths in our paper. We acknowledge the limitations you have listed and plan on posting a revised manuscript in the coming days to help address your concerns. Until then, we have a few clarifying questions. First, when you request an evaluation of “the impact of normalization on the metrics,” do you mean “regularization” in the case of CCA-based metrics? If not, can you clarify which “normalization” you are referring to in this context? Second, what additional details would you like to see to help clarify how “performance deficits were computed?” In our original manuscript, we wrote the following:
>
> >We recorded the average classification rank of the target class. For instance, if the CNN's softmax layer predicted that the Junco bird class was the third most likely class given an image of a Junco bird, then that image received a rank of three. The average class-specific ranks intuitively measure how functionally useful the representations from the representation matrix were to the network. We refer to the changes induced in class-specific classification rank by ablation as network performance deficits.
>
> In response to reviewer feedback, we have adapted our measure of network performance deficits to use the logit from the softmax layer instead, resulting in the updated description:
>
> > Following a procedure similar to [1], we first record the logits from the final layer of the network. We then normalize the logits using a standard z-score such that each logit entry equals $\hat{\pi_i} = (\pi_i - \mu(\pi)) / \sigma(\pi)$. Next, we calculate the effect of the ablation on the classification task by computing the difference between the normalized logit for the target class with and without ablation, $\Delta_\text{ablate} = [ \pi_t(x) - \pi(x | \text{do}(z)) ]$, where $x$ is the input image, $\pi_t$ is normalized logit for the target class, and do$(z)$ represents the ablation on the hidden representation $z$. We refer to the changes induced in class-specific classification performance by ablation as network performance deficits.
>
> >[1] McGrath, T., Rahtz, M., Kramar, J., Mikulik, V., & Legg, S. (2023). The hydra effect: Emergent self-repair in language model computations. arXiv preprint arXiv:2307.15771.
>
> What additional details would help clarify the methodology?

---

> > ### Comment · Reviewer_x8XN · 2024-01-17
> >
> > Thank you for following up. Yes, I mean the regularization. For the performance deficits, what metric did you use to compare the predicted and ground truth ranks?
> >
> > Apart from that, it would be great if you could add more details on the statistical testing and say how what you are measuring shows causal effects.

---

> ### Author Response · Authors · 2024-01-24
> **Revised Manuscript Upload (Part 1)**
>
> Thank you very much for your thoughtful and constructive feedback. Below we address your concerns individually.
>
> > **_It was not clear to me how the changes the authors are inducing in the networks are causal and to what extent the tests that the authors use are causal. This deserves more discussion as it is the central claim of the paper._**
>
> In the original manuscript, we called the changes induced in the networks and the associated tests "causal" to reflect the fact that we performed interventions (zero ablations) on representations of the network. Interventions on a representation $X \in \mathbb{R^{m \times n}}$ come in the form of zero ablations where entire features (columns of $X$) are set to zero using a mask producing an ablated representation $A \in \mathbb{R^{m \times n}}$ where $A = M \otimes X$. In the updated manuscript, we represent the zero ablation intervention on functionality with the causal do$(\cdot)$ operator.
>
> However, after considering reviewer feedback, we would like to transition away from the use of the word "causal" in this context. Interventions like those employed in this study are typically used to infer the causal role of representations (McGrath et al. (2023)). Here, we do not aim to test the causal role of representations directly. Instead, we use interventions on representations to test RSMs for their ability to infer functional similarity from representational similarity. In the context of the current study, our interventions are not causal because they are not aimed at revealing causal properties of representations. For this reason, we omit the term "causal" in our revised manuscript.
>
> Similarly, we avoid the use of the term "non-causal" in the revised manuscript. In our original manuscript, non-causal tests were those that involved processing representations using linear decoders rather than the downstream network. We originally called them "non-causal" tests because interventions that affect linear decoders do not necessarily "cause" changes in network performance. After reflecting on the terminology, we have concluded that the term "non-causal" is inappropriate because we do not study causality specifically using these tests. Instead, we use linear probes as a measure of class separability in the representation. Please refer to the updated manuscript introduction for more details.
>
> > **_For the presentation of results, it would be great to see the actual similarity values in addition to the correlation coefficients. In particular, it is unclear whether the similarity values are what you would expect at the moment, i.e. more similar networks have higher similarity scores. There could be a good correlation even if the actual values do not make sense, so it is important to show them._**
>
> Thank you very much for this feedback. We would love to be able to interpret the similarity values from our study. One "expectation" we have for RSM scores is that they should increase (i.e. the representations should be considered less similar) when ablations produce larger network performance deficits. This is exactly what we test for using our correlation analysis. If we achieved a perfect correlation between network performance and CKA distance, for example, then it would be possible to map arbitrary CKA values to changes in network performance. However, without any grounding in interpretable metrics, RSM scores are notoriously difficult to interpret (Raghu et al. (2021)).
>
> > **_The statistical tests the authors run are unclear. The authors say that they compute the Spearman correlation (rank?) coefficient, but also report p-values -- how were those computed? Did you correct for multiple testing?_**
>
> Thank you very much for this feedback. We have overhauled Section 3 with more details that will hopefully make the process clear. To address your question, we compute Spearman rank correlation coefficients and perform a hierarchical linear model to evaluate the prediction of the rank correlation values as a function of two primary factors: Metric and Functionality, which is how we obtain p-values.

---

> ### Author Response · Authors · 2024-01-24
> **Revised Manuscript Upload (Part 2)**
>
> > **_Similarly, it was unclear to me how the performance deficits were produced. For the performance deficits, what metric did you use to compare the predicted and ground truth ranks?_**
>
> In your question it seems like you are referring to three different ranks: ground truth, predicted, and predicted after ablation. The ground truth ranks are always 1. In other words, if Junco Bird is the ground truth label for an image, the model should predict that the Junco Bird class is the 1st most likely class for that image. However, the models aren't perfect and will sometimes predict that the Junco Bird class is the 2nd, 3rd, 4th, Nth most likely class for a Junco Bird image. After an ablation, if the ablation hurts performance, then the model should predict that the target class is the Mth most likely class where M > N. In that case the performance deficit given an ablation on the representation is M - N. In a previous comment, we mentioned that had changed our performance deficit calculation to use the logit. However, we ran out of time to include these new results in the updated manuscript.
>
> > **_What is the impact of regularization on the CCA-based metrics? The authors note that they perform worse than CKA and Procrustes; could this be a consequence of the normalization?_**
>
> This is an excellent question also raised by another reviewer. We choose regularized versions of CCA to estimate representational similarities in our analysis for computational reasons outlined in the manuscript. There is some concern this estimation could introduce errors into our results that interfere with the conclusions of the paper. We have run additional experiments to test different penalties for regularizing the CCA metrics. We find that while the exact CCA values change when the penalties change, their relative orders do not. In the current study, we are primarily concerned with the relative order of scores given an ablation. We find that CCA metrics regularized with different penalties are highly correlated (Spearman rank correlation > 0.95) and therefore the chosen penalty does not interfere with the main results. Because regularization is required to perform CCA-based metrics on matrices $X \in \mathbb{R^{m \times n}}$ where $m < n$, we cannot directly compare unregularized and regularized metrics. In future work, we would like to expand on these results to establish best practices for computing RSM scores on large layers.
>
> > **_For Figure 4, please use the same scale for each sub-plot -- it looks like the differences for MobileNetV2 are largest, but this is just an artifact of the different scale._**
>
> Thank you for pointing this out. We have updated the figures in the revised manuscript.

---

> > ### Comment · Reviewer_x8XN · 2024-01-26
> >
> > Thank you for the changes and clarifications. It is concerning that you were not able to include the new results for the logit deficit calculations after you decided that these make more sense.

---

> > > ### Author Response · Authors · 2024-01-29
> > >
> > > We agree that using the logit could be beneficial for comparing our results to previous studies. We had used the logit as part of a new set of analyses we ran in response to another reviewer’s comments. Because the rank deficit score is a discretized version of the logit score and obtaining the logit scores would require all computations to be repeated, we chose to report the deficit calculation as the rank-deficit.

---

### Review · Reviewer_vUVK · 2024-01-09

**Summary Of Contributions:**

The paper systematically tests representation similarity metrics to evaluate their sensitivity to causal functional changes induced by ablation. The authors use changes in network performance after ablation as a way to causally measure the influence of representation on function. They compare this to changes in linear decodability, which measures the available information in the representation.

The results show that all tested metrics are more sensitive to decodable features than network performance. Among the metrics, Procrustes and Centered Kernel Alignment (CKA) outperform regularized Canonical Correlation Analysis (CCA)-based methods on average. However, these metrics have a diminished advantage when considering network performance.

The paper concludes that interpretability methods will be more effective if they are based on representational similarity metrics that have been evaluated using causal tests. They also suggest that further tests are needed to evaluate representation similarity metrics and to understand what important pieces of information similar representations share.

**Audience:**

Yes

**Claims And Evidence:**

Yes

**Requested Changes:**

Include results with non-vision models and with larger problems (more classes)

**Strengths And Weaknesses:**

## Strengths

1. Well written and easy to follow paper

2. Systematic comparison of representation similarity metrics

3. To the best of my knowledge, these results are novel.

## Limitations

1. The paper is primarily focused on vision models (AlexNet, MobileNet, or ResNet50). Their results may not generalize to other models.

2 The scale of problem (in terms of the number of classes) is quite small.

---

> ### Author Response · Authors · 2024-01-17
> **Thank you and clarification on requested changes**
>
> Thank you very much for your thoughtful review and for recognizing strengths in our paper. We acknowledge the limitations you have listed. First, we agree that the results shown in this paper for convolutional networks trained on ImageNet may not generalize to other network architectures and tasks. As you have pointed out, increasing the number of classes analyzed will help build evidence to support our claims. In the updated manuscript, which we plan on posting in the coming days, we hope to include results for 50 classes on all networks. However, we do not have the time necessary to include results for other networks architectures or tasks. That being said, we will restrict the claims we make in the revised manuscript to reflect only the experiments we perform in the paper. Does this help address your concerns? Is there something else you can imagine might help mitigate your concerns that we could accomplish in the available time?

---

> ### Author Response · Authors · 2024-01-24
> **Revised Manuscript Upload**
>
> Thank you very much for your thoughtful review and for recognizing strengths in our manuscript. We also recognize the limitations of our work and address your concerns below.
>
> > **_The paper is primarily focused on vision models (AlexNet, MobileNet, or ResNet50). Their results may not generalize to other models._**
>
> We chose our three networks as a representative sample of three convolutional networks with different architectures that are still widely studied for producing, testing, and analyzing representation similarity metrics (Kornblith et al. (2019); Raghu et al. (2021); Boix-Adsera et al. (2022)). We acknowledge the fact that our results might not generalize to other models or tasks. Accordingly, we have edited our manuscript to better limit the claims to reflect the scope of the current study. In future works we hope to expand our analysis to other networks and tasks.
>
> > **_The scale of problem (in terms of the number of classes) is quite small._**
>
> We chose 50 classes for AlexNet and 10 classes for MobileNet and ResNet to strike a balance between convenience and breadth. Computing RSM scores requires working with a representation matrix $X \in \mathbb{R}^{m \times n}$, where $m$ is the number of images and $n$ is the number of neurons in a given layer. Given the large layer sizes of MobileNet and ResNet (~100,000 neurons), adding more classes increases the cost and memory requirements of computing RSM scores. To help avoid bias in our small sample of classes, we sample the classes completely randomly from ImageNet. We acknowledge that increasing the number of images is desirable and we plan on including more images in future work, however, we note that previous studies include a comparable number of images in their analysis (Raghu et al. (2021)).

---

### Decision · Action_Editor_KnuM · 2024-03-12

**Recommendation:** Accept with minor revision

**Comment:**

The paper were reviewed by three reviewers. After rebuttal and revision, two reviewers were positive (Accept and Leaning accept), while one reviewer was negative (Leaning reject).

Reviewer DRWf wrote a very detailed review. This reviewer was initially concerned with the wordings about "causal" and "non-causal", but was satisfied with the authors' revision with improved and clarified terminology. Other suggestions by the reviewer in terms of the presentation were also incorporated in the revision. In the end, Reviewer DRWf was positive about the revision.

Reviewer vUVK was concerned with the restriction to the vision models and the limited scale of the experiments. The authors addressed the concerns in the rebuttal.

Reviewer x8XN was negative after rebuttal, primary because the revision did not include results for the logit deficit calculations. The authors reported rank deficit calculations, which can be considered discretized versions of logit deficit calculations. This reviewer was also concerned with the wording "causal" in the title, but it has since been removed from the title.

Overall, I feel that the paper could be made more substantive by enlarging the scope of the experiments. However, I concur with Reviewer DRWf and Reviewer vUVK that the current version is acceptable.

**Audience:**

The paper can be of interest to deep learning researchers and practitioners who care about understanding and interpreting the learned neural networks.

**Claims And Evidence:**

The paper claims that representation similarly may not capture function similarity between neural networks. The claim is supported by ablation studies that show that representation similarity metrics are more sensitive to decodable features than network performance. The scope of the experiments is limited in terms of the size of the dataset and the types of network architectures.

---

> ### Author Response · Authors · 2024-04-04
> **Response to Decision by Action Editor KnuM**
>
> Dear Action Editor KnuM,
>
> We greatly appreciate all of the time and effort you have put into helping us improve this manuscript. Your choice of reviewers helped the paper (and our thinking) evolve in ways we believe will be beneficial both to our future work and hopefully the field more broadly.
>
> Based on your post, we have prepared a camera-ready version of the last version of the paper we posted for review. Before we post this new version, is there anything else you would like to see edited or added to the paper?